



# Observations of Electric Fields during two Partial Solar Eclipses at the Geomagnetic Equator

Manuel A. Bravo[1], Joel H. Fernández[2], Adán Godoy[1], Jackson E. Pérez[2], Benjamín A. Urra[1], Antonela Ore[2], Enrique A. Carrasco[1], Juan J. Soria[2], Enrique D. Rojo[1], Carlos E. Saavedra[2], Elías M. Ovalle[1],

Sulamita M. Ramos[2], Helen C. Meza[2], Giancarlo E. Mayhuire[2], Pedro Quispe[3], Eduardo Vigo[3], Orlando Poma[2]

[1]Centro de Instrumentación Científica, Universidad Adventista de Chile, Chillán, 3780000, Chile

[2]Escuela Profesional de Ingeniería Ambiental, Universidad Peruana Unión, Lima, 150118, Perú.

[3]Escuela Profesional de Ingeniería Ambiental, Universidad Peruana Unión, Juliaca, 21101, Perú,

*Correspondence to*: Manuel A. Bravo (manuelbravo@unach.cl), Orlando Poma (opoma@upeu.edu.pe)

**Abstract.**

This study presents the first coordinated observations of atmospheric electric field (AEF) and ionospheric plasma drifts during the partial solar eclipses of 2 July 2019 and 14 October 2023, observed near the magnetic equator in Lima, Peru. AEF was measured using a field mill, while ionospheric drifts were obtained from radar observations at the Jicamarca Radio

Observatory and local magnetometers. The two events displayed contrasting electrodynamic responses: in 2019, AEF variations were ambiguous due to meteorological fluctuations, while in 2023, clearer weather conditions revealed distinct decreases in both surface AEF and ionospheric vertical drift near maximum obscuration. These results demonstrate the variable nature of eclipse-time electrodynamics and emphasize the importance of multi-instrument approaches for understanding atmosphere-ionosphere coupling in low-latitude regions.

## 1 Introduction


Solar eclipses have long captivated humanity, not only for their striking visual spectacle but also for the physical changes they induce in Earth's environment. Beyond partially or completely blocking sunlight, the Moon alters various atmospheric and geophysical parameters. Some of these effects are well documented, such as reductions in radiation and temperature (e.g., Zerefos et al., 2000; Peñaloza-Murillo and Pasachoff, 2018; Calamas et al., 2019), as well as variations in

meteorological factors like wind, pressure, and relative humidity (e.g., Anderson et al., 1970; Winkler et al., 2001; Lazzús et al., 2022). However, solar eclipses also influence Earth's electric and magnetic fields, an area that remains less understood despite numerous studies exploring their impact.



Regarding the vertical atmospheric electric field (AEF), research findings have been inconsistent, with discrepancies between experimental data and theoretical interpretations. For instance, Anderson and Dolezalek (1972) observed a brief
increase in electric field at ground level following totality during the eclipse of 7 March 1970. Similarly, Babakhanov et al. (2013) recorded a sharp increase in AEF at the peak of the 1 August 2008 total eclipse in Novosibirsk, Russia, preceded by a transition from negative to positive values. A more pronounced increase of potential gradient for the same eclipse is measured in Kolkata, India (De et al., 2010) and is like that observed in the VLF measurements. In contrast, Kumar et al. (2013), studying the annular eclipse of 15 January 2010, reported a significant drop (up to 65%) in AEF, with the AEF
showing periods of enhancement both during and after the eclipse until sunset in the Indian sector. Bennett (2016) found no effects of the eclipse of 20 March 2015 on the atmospheric electric field in the UK, with variability during the eclipse being comparable to pre- and post-eclipse conditions. Likewise, Tacza et al. (2016) recorded a ~55 V/m increase in AEF at two detectors, 0.4 km apart, during the total eclipse of 11 July 2010 at the Complejo Astronómico El Leoncito (CASLEO). They proposed a possible link between the lower ionosphere and the lower atmosphere, as similar variations were observed in very
low-frequency (VLF) signals during the eclipse.

Even Dhanorkar et al. (1989), for the solar eclipse of March 18, 1988, detected changes in the electric field well before sunrise (and therefore before the eclipse occurred in Pune, India). Others have observed changes in the electric field up to 3-4 hours after the eclipse of 16 February 1980 over Indian Region (Manohar et al., 1985).

The AEF, however, is also influenced by local meteorological factors such as altitude, latitude, temperature, wind, and
humidity. That is, eclipse-induced changes in air conductivity can affect AEF measurements. Considering this, Velazquez et al. (2022) examined atmospheric electrical and meteorological variations during the total eclipse of 14 December 2020 at three locations in Argentina: Valcheta (100% eclipse), Buenos Aires (73%), and CASLEO (71%). Despite the eclipse reducing solar irradiance, no clear effects on the near-surface AEF were observed, likely due to local weather conditions. Notably, at Valcheta, near a frontal zone with clouds and dust, AEF values were significantly higher and opposite to typical
fair-weather conditions. Meanwhile, in Buenos Aires and CASLEO, AEF values during the eclipse were more consistent with fair-weather behavior, though a slight decrease in AEF was detected.

As for the magnetic field, Vega-Jorquera et al. (2021) reported a synchronized increase of ~12 nT in all components of a fluxgate magnetometer during the total eclipse of 2 July 2019, indicating an overall strengthening of the geomagnetic field. Conversely, Liu et al. (2022) observed a weakening of all magnetic field components during the annular eclipse of 21 June
2020. Similarly, Meza et al. (2021) detected reductions of up to 6 nT in all magnetic field components at stations in Argentina during the total eclipse of 14 December 2020. In agreement with these findings, Chen et al. (2023) observed even larger reductions (10–15 nT) at low-latitude stations during the same eclipse. However, in some cases, such as the study by Babakhanov et al. (2013) on the 1 August 2008 total eclipse, magnetic field effects were less apparent and only became evident when compared to measurements from other observatories.





This study focuses on the relationship between the electric field and solar eclipses. On a global scale, Earth can be conceptualized as a vast capacitor, with the upper ionosphere and the planet's surface acting as its plates, separated by the dielectric atmosphere. Within this dielectric, an atmospheric electric field exists, which can be influenced by solar eclipses, particularly when significant voltage differences occur between the plates (Martínez, 2014).

Given the coupling between the lower ionosphere and the lower atmosphere during solar eclipses, the electric field measured
at the Earth's surface should, to some extent, reflect disturbances occurring at ionospheric altitudes. Specifically, the electromagnetic E×B drift measured at Jicamarca (12.0°S, 76.9°W; dip ~1°N) offers valuable insight into ionospheric electric field variations during solar eclipses due to its unique equatorial geomagnetic conditions (see St-Maurice et al., 2011; Chen et al., 2023, among others). This study aims to compare and analyse the effects of electric fields in both the lower atmosphere and the ionosphere during two eclipses: the total solar eclipse of 2 July 2019 and the annular eclipse of 14
October 2023. Given the path of these eclipses partially through equatorial regions, they provide a unique opportunity to study ionospheric, electrodynamic, and magnetic variations at low latitudes during solar eclipses (Ouar et al., 2024).

## 2 Data and methodology

*Solar eclipses*

The total solar eclipse on 2 July 2019, was visible across much of South America (see Figure 1). The path of totality began
over the Pacific Ocean, crossing the continent and starting near La Serena, Chile (29.9°S, 71.3°W) at 20:38 UT, and ending near Buenos Aires, Argentina (34.6°S, 58.3°W) at 20:45 UT, just before sunset. The eclipse occurred during an extended period of quiet geomagnetic activity conditions and very low solar activity (15 quiet days between June 29 and July 20 with Kp ≤ 2+ and the observed F10.7 index ≤70, Bravo et al., 2020). In Jicamarca, the eclipse at 300 km altitude began at 19:27 UT until 21:53 UT, passing through a maximum of 56% obscuration at 20:46 UT (Bravo et al., 2020).

The annular solar eclipse on 14 October 2023 began with the penumbra's arrival in the northwest of North America at 15:03 UT (41°N, 132°W), followed by the umbra's arrival at 16:10 UT (48°N, 146°W) (see Figure 1). The eclipse then moved southeastward, crossing through Central and South America, before concluding at 19:49/20:55 UT (umbra/penumbra) over the Atlantic Ocean (6°S, 29°W) and to the east of Brazil (13°S, 45°W). Such a broad latitudinal path across the Americas is highly unusual (Ouar et al., 2024). In the case of Jicamarca, the eclipse at 300 km altitude began at 17:31 UT until 20:38 UT,
passing through a maximum of 54% obscuration at 19:09 UT.




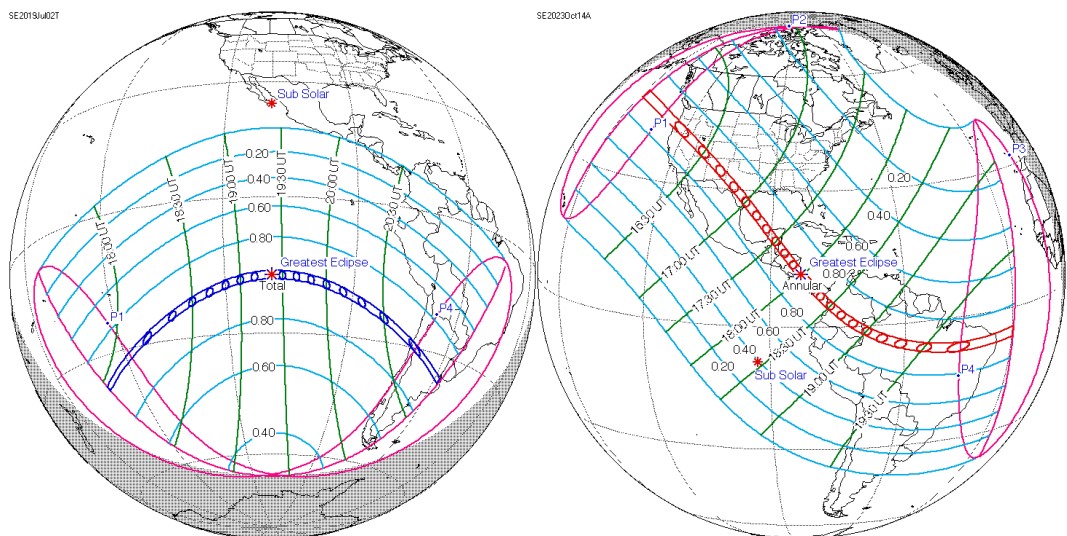

**Figure 1: Solar eclipse paths on (left) 02 July 2019 (right) 14 October 2023 (source: https://eclipse.gsfc.nasa.gov/).**

*Atmospheric electric field and meteorological data*

Atmospheric electric field measurements were recorded using a commercially available Electric Field Mill (EFM) manufactured by Boltek Corporation (model EFM100-1000120-050205). The EFM has a dynamic range of ±20 kV/m. Its operation is based on fundamental electromagnetic principles: when a conducting plate is exposed to an electric field, a charge is induced that is proportional to both the electric field and the plate's area (Tacza et al., 2016). The sensor is installed in Universidad Peruana Union (UPeU) campus Lima (12.0 °S; 76.8°W). The electric field measurements are taken with a time resolution of 0.05 seconds and then integrated using 1-minute averages for the analysis presented. The sensor is part of a collaboration that involves the AFINSA network (https://theafinsa.wordpress.com/instrumentation/), an organization focused on electrical field research, in partnership with the Aerospace Research Agency (CONIDA). This partnership is significant due to the extensive network of sensors that AFINSA has distributed throughout Latin America. The time series spans from December 2018 to September 2024, though it contains significant gaps due to maintenance, power outages, and other operational interruptions. Additionally, the initial period (2018–2021) exhibits high dispersion in the data, primarily caused by the lack of post-installation height correction for the sensor. This issue was resolved by late 2018. Given these factors, the atmospheric electric field (AEF) data are presented in normalized form, shifting the analytical focus toward visual pattern comparison rather than absolute amplitude values

Meteorological data were collected using an industrial-grade Davis Instruments Vantage Pro2 weather station, co-located with the electric field sensor. The station records measurements at a high temporal resolution (2.5-second intervals). For this study, raw data were averaged over 5-minute intervals to mitigate high-frequency fluctuations while preserving relevant atmospheric variability. The dataset covers the period from November 2019 to September 2024, and it is used to select control days for the electric field considering quiet meteorological variable conditions (see Soria et al., 2021).



Two different approaches were used to define the Fair Weather (FW) curve for both eclipses. For the July 2019 eclipse, in the absence of a meteorological station on site, a statistical criterion was applied based on the methodology employed by

Velázquez et al. (2024) and Lucas et al. (2017), which uses a robust statistical approach involving the calculation of the median of the time series and its median absolute deviation (MAD). Outliers were excluded by applying a threshold of ±5 MAD relative to the median, thereby removing significant meteorological and instrumental disturbances. Days meeting this criterion were classified as "Fair Weather" (FW) days. From these, a reference curve was constructed as the average of the selected days. Due to the high data dispersion, associated with the lack of sensor height calibration during its initial

operational period, the results were normalized following the method proposed by Tacza (2018). The reference curve was calculated using 442 days of data, averaged at 1-minute intervals.

For the October 2023 eclipse, a meteorological-based approach was adopted (Velázquez, 2021). Fair-weather days were defined as those satisfying three specific conditions: (i) relative humidity below 95%, to exclude precipitation events that could alter the local electric field; (ii) hourly average winds below 8 m/s, to minimize charge drag effects, (iii) a Pearson

correlation coefficient between the observed radiation and a simplified theoretical curve greater than 0.95, indicative of clear skies. Applying these criteria, 139 days were identified under FW conditions.

Data from the onboard Advanced Baseline Imager (ABI) on geostationary satellite GOES-16 (Geostationary Operational Environmental Satellite) are used to observe the weather. Only band 13 (10.1 to 10.6 µm bandwidth), and spatial resolution of 2 km, was processed following Huamán 2025 GOES package's tutorial, only adding the map projection PlateCarree and a

special color palette for Infrared Channel Enhancement from Rojas 2021.

*Vertical drift data*

Vertical and zonal drift data E×B is obtained from Madrigal Database (https://www.igp.gob.pe/observatorios/radio-observatorio-jicamarca/madrigal/) at Jicamarca Radio Observatory, Perú. Specifically, the 2019 eclipse period corresponds to data measured by the Jicamarca Incoherent Scatter Radar (ISR), and the 2023 eclipse period to data measured by the

JULIA-MP (Jicamarca Unattended Long-term Investigations of the Ionosphere and Atmosphere-Medium Power) radar mode (see Kuyeng et al., 2023). The data correspond to the averages of vertical drifts between 247 and 546 km, with steps of 60 km, measured in units of m/s. Values closest to 300 km were used.

Magnetometer data from the Huancayo Geomagnetic Observatory (HUA), located near the EFM, were utilized to estimate the vertical drift. This was achieved by subtracting the H component recorded at the Arequipa Magnetic Station (ARQ), part

of the LISN network (http://lisn.igp.gob.pe), from the HUA data. The median values during nighttime were removed, and a polynomial model was applied to convert the results into equivalent E×B values, using values of Kp, Ap and F10.7 (Anderson et al., 2002). These indexes were obtained from the OMNI Web (https://omniweb.gsfc.nasa.gov).





# 3 Results

*Atmospheric electric field (AEF)*

Observations of the atmospheric electric field measured during both solar eclipses are presented in Figure 2. They are compared with other nearby days. Although there is high variability, at first glance it appears that in both situations there is an increase in the time of maximum obscuration (vertical black continuous line).

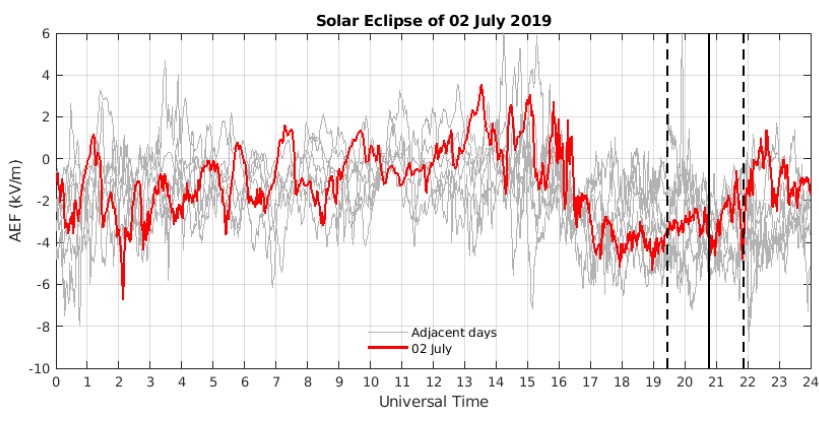

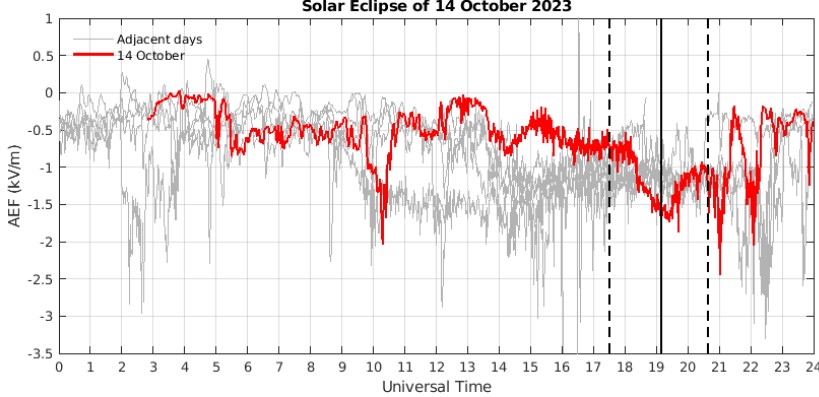

**Figure 2: Atmospheric electric field measurements during the day of the solar eclipse and the surrounding days. (top) 02 July**
**2019, (bottom) 14 October 2023 solar eclipses. Maximum obscuration (vertical black continuous line), eclipse onset and eclipse end (vertical black dashed line) are indicated.**

The top panel of Figure 2 shows the atmospheric electric field for the 2 July 2019 eclipse, along with the three days before and three days after. On the other hand, for the 14 October 2023 eclipse (bottom panel), it is shown only along with the day





of 11 October 2023 and three days after. This is due to equipment failures and/or power outages. In general, there are few
continuous periods of data.

Due to the high day-to-day variability of the atmospheric electric field, influenced by various factors, it is necessary to
establish a control curve through the FW analysis. Figure 3 and Figure 4 show the FW curves for the eclipses of 2 July 2019,
and 14 October 2024, respectively.

Fair-Weather curve according to statistical criterium (FW-SC) in Figure 3 shows that the overall trend of the FW-SC curve is

upward during the eclipse, though with a sharp decline after maximum occultation. The normalized standard deviation (gray
area) reveals significant dispersion, attributable to the lack of sensor calibration during its initial installation (2018–2021). It
is important to note that during this eclipse, no meteorological parameters were measured at the station. However, the GOES
image shows no significant severe weather activity at the time of maximum obscuration (20:50 UT).

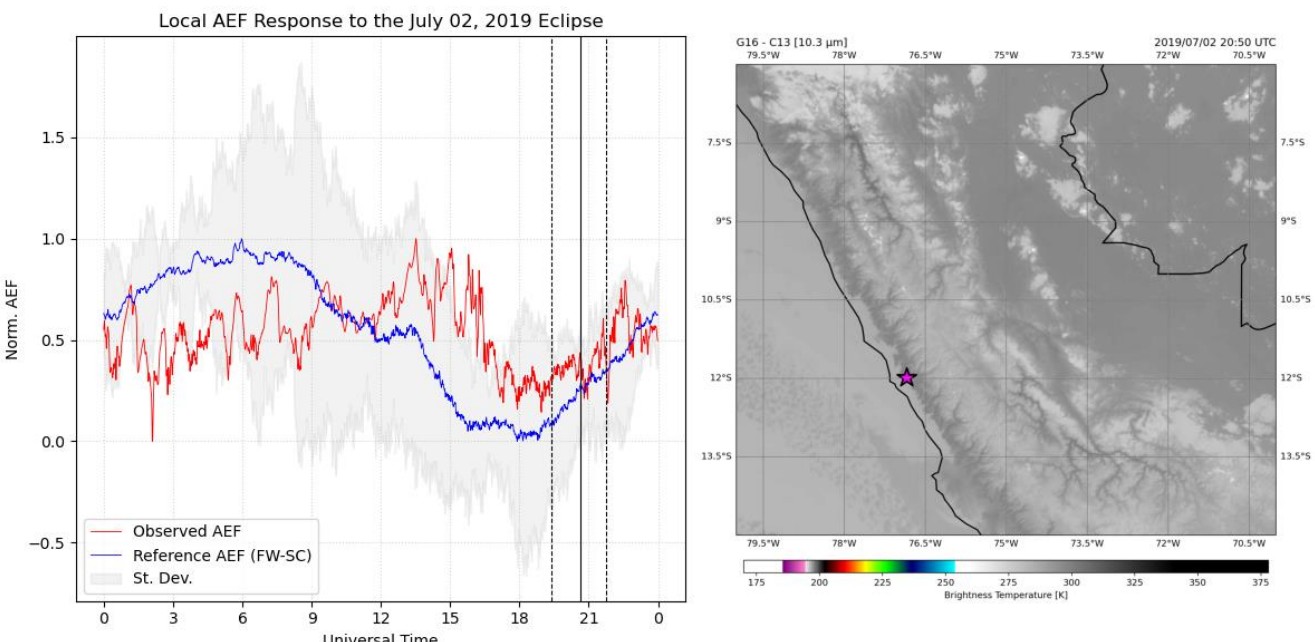

**Figure 3: (left) Fair Weather curve for atmospheric electric field measurements during the day of the solar eclipse on 2 July 2019**
**according to statistical criteria (FW-SC). Maxima obscuration (continous vertical black line), eclipse onset and eclipse end (dashed**
**vertical black lines) are also indicated. (right) GOES-16 map at 10.3 μm. The magenta star shows the location of the sensor.**

On the other hand, like Velazquez et al. (2022), Figure 4 presents the parameters of solar radiation (Rad), temperature (T)
and relative humidity (RH) accompanied by the observed AEF and FW curves according to two criteria for the 14 October

2023 eclipse. FW-SC curve is the same as that presented in Figure 3 and the fair-weather curve according to the
meteorological criterion (FW-MC) is also presented. Both reference curves exhibit a high correlation ($r = 0.97$), validating



the robustness of the statistical criterion. The most notable differences occur between 14:00 and 18:00 UTC, with higher values in the FW-MC curve.

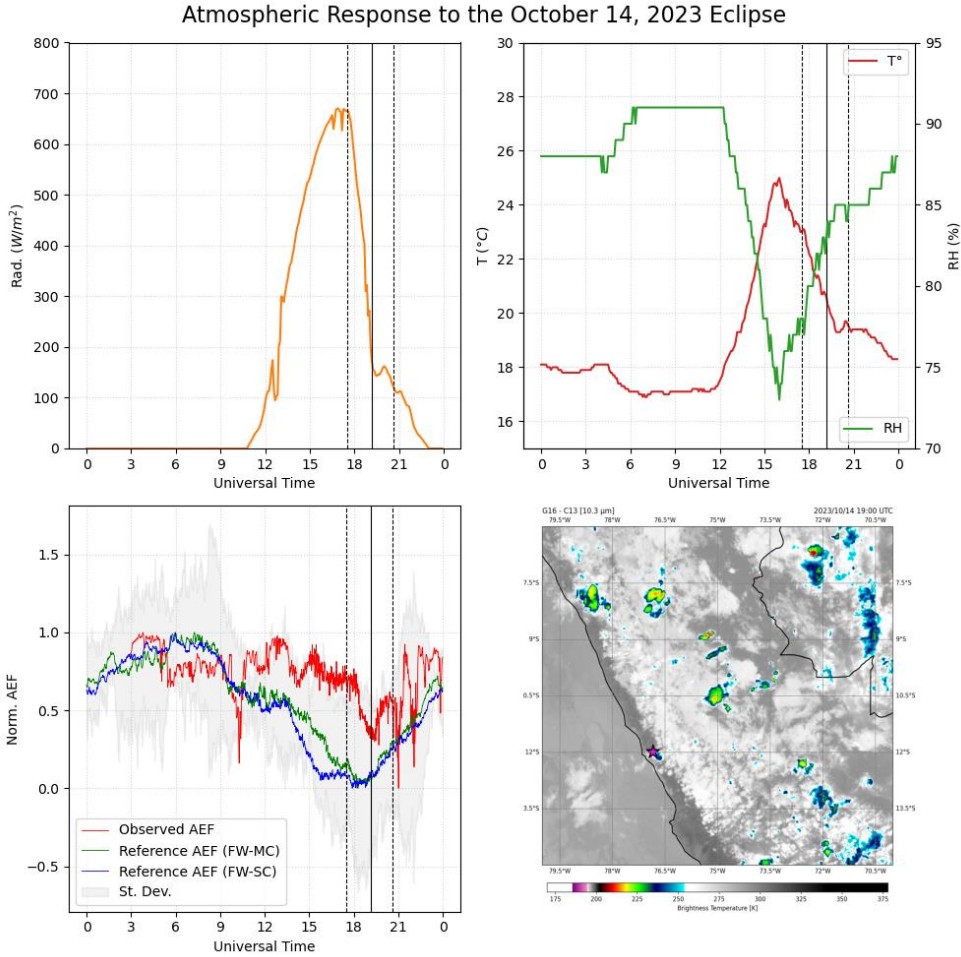

**Figure 4: (Top panels) Meteorological variables recorded during the eclipse day: solar radiation (Rad), relative humidity (RH), and temperature (T). (Bottom panels) Observed atmospheric electric field (AEF) during the event, alongside the two reference curves derived from statistical (FW-SC) and meteorological (FW-MC) criteria. Maximum obscuration (continuous vertical black line), eclipse onset and eclipse end (dashed vertical black lines) are indicated. GOES-16 map at 10.3 μm. The magenta star shows the location of the sensor.**

During the eclipse, solar radiation followed a Gaussian profile, consistent with clear-sky conditions, as confirmed by satellite images showing minimal cloud cover near the sensor location. Radiation decreased abruptly after the eclipse onset, reaching a post-occultation minimum followed by a slight recovery before the end, and a subsequent decline (approaching the terminator). This pattern was replicated in temperature. Relative humidity showed no abrupt variations (daily maximum:



92%), ruling out precipitation. Wind (figure not shown) recorded its daily peak before the eclipse (11 m/s), decreasing to 4
m/s by the event's end.

The atmospheric electric field (AEF) responded to the eclipse with a temporal delay: its decrease began shortly after the
event started, and its recovery coincided with that of radiation. In both eclipses, the AEF diverged from the reference curves,
exhibiting rapid fluctuations.

The GOES image depicts a small cold cloud over the station (magenta star), with moderate meteorological activity observed
in areas located over 200 km away at the time of maximum obscuration (~19:00 UT).

In summary, comparing the FW curves for both eclipses makes it challenging to clearly identify the effect of the eclipse,
even though a potential influence is noticeable to the naked eye. This uncertainty may be attributed to the high variability of
the atmospheric electric field, which is influenced by multiple environmental and meteorological factors.

*Ionospheric electric field*

Figures 5 and 6 show the observations of the E×B drift during both eclipses, both the vertical drift deduced with
magnetometers and the vertical and zonal velocities measured at Jicamarca (approximately 5.5 km from the EFM sensor
location). Since the drift derived from magnetometers is considered representative only during daylight hours, the graphs
only show daylight hours, between 06:00 and 19:00 LT (11:00-24:00 UT). The right-hand panels show the comparison
between the reference day and the day of the solar eclipse. Due to Jicamarca radar measurements are made during
campaigns, the nearest geomagnetically quiet day where measurements exist was considered the reference day. These days
are July 3, 2019 (Ap=5) for the July 2019 eclipse, and October 12, 2023 (Ap=4). The right-hand panels show the differences
between the eclipse days and the reference days.





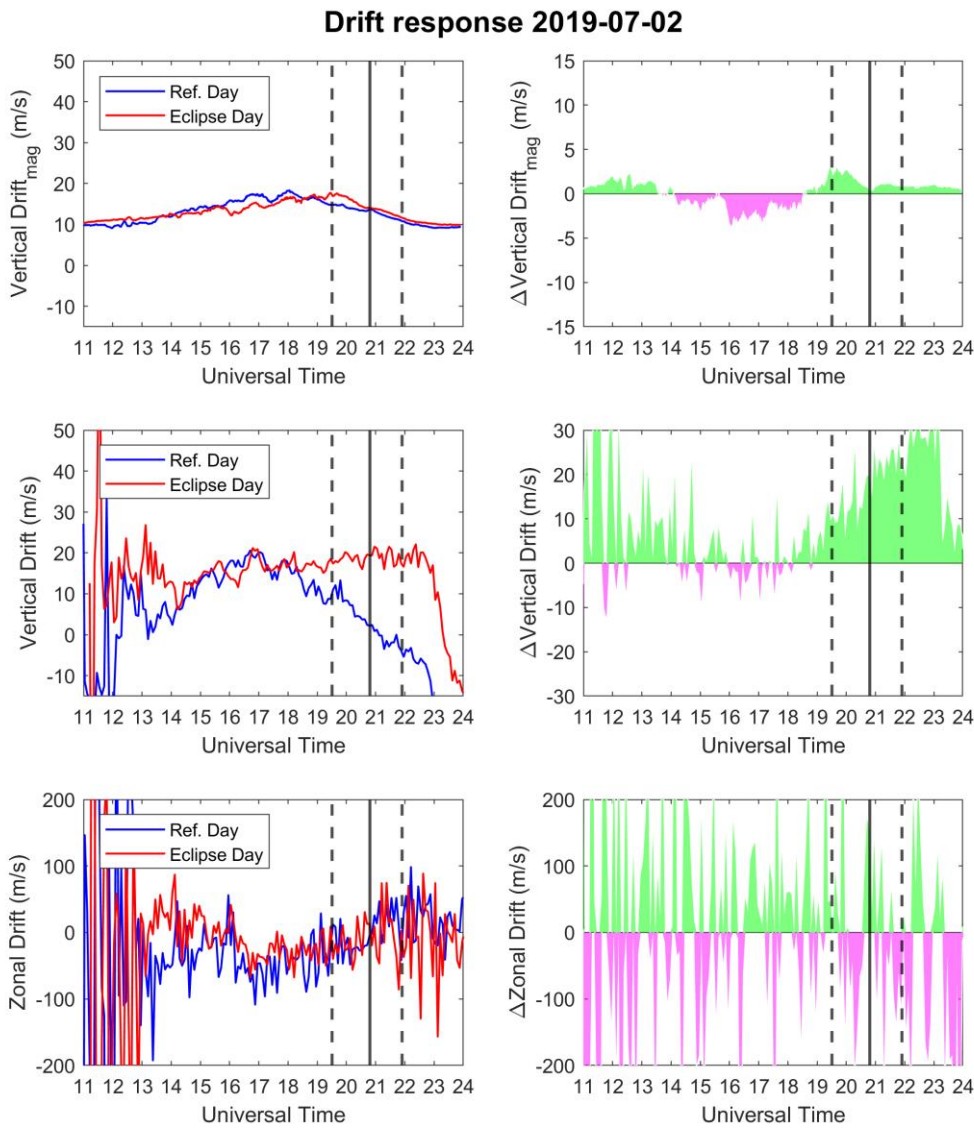

**Figure 5: Vertical and zonal drift E×B values (left) during the solar eclipse on 2 July 2019 and control day (3 July 2019).**
**Differences between the eclipse day and the reference (right) are shown, where positive values are colored green and negative**
**values in magenta. Maximum obscuration (vertical black continuous line), eclipse onset and eclipse end (vertical black dashed line)**
**are indicated.**



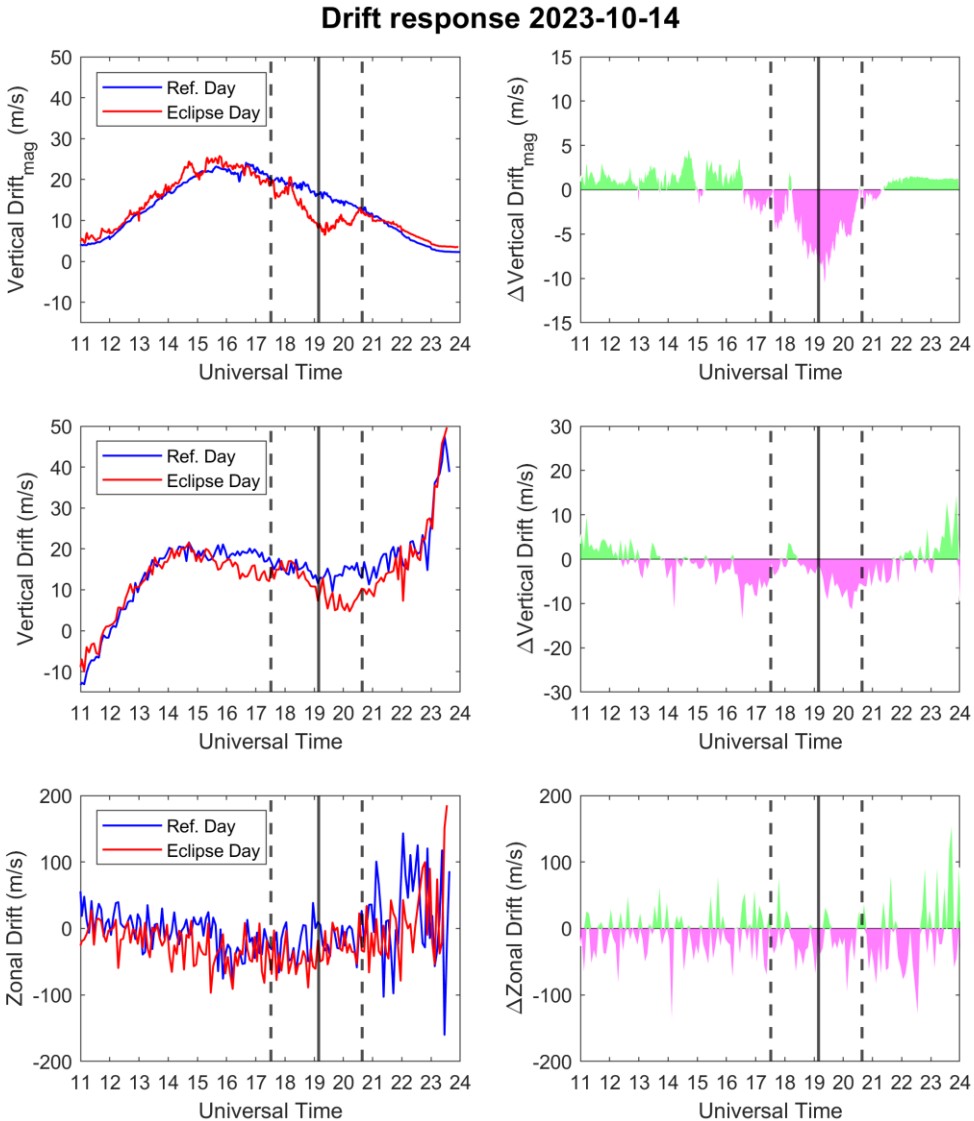

**Figure 6: Vertical drift E×B values (left) during the solar eclipse on 14 October 2023 and control day (12 October 2023). Differences between the eclipse day and the reference (right) are shown, where positive values are coloured green and negative values in magenta. Maximum obscuration (vertical black continuous line), eclipse onset and eclipse end (vertical black dashed line) are indicated.**

Each eclipse event presents a distinct response. For the 2019 eclipse, a positive effect on the vertical drift is observed, while for the 2023 eclipse, the effect is negative. Particularly for the 2019 eclipse, a very slight increase in the vertical drift




deduced from magnetometers is observed, while the measured vertical drift is very significant during and after the eclipse. For the 2023 eclipse, the decrease is similar for both vertical drifts, except that the measured one presents a phase shift with respect to maximum obscuration. In both cases, the change in zonal drift is less evident or very unclear.

The vertical drift velocities measured at Jicamarca, representing averages within the 247 to 546 km altitude range, can be utilized to estimate the zonal electric field ($E_y$). Since the E×B drift velocity is equal to E/B, a vertical drift of approximately 40 m/s corresponds to a zonal electric field of 1 mV/m (Anderson et al., 2004).

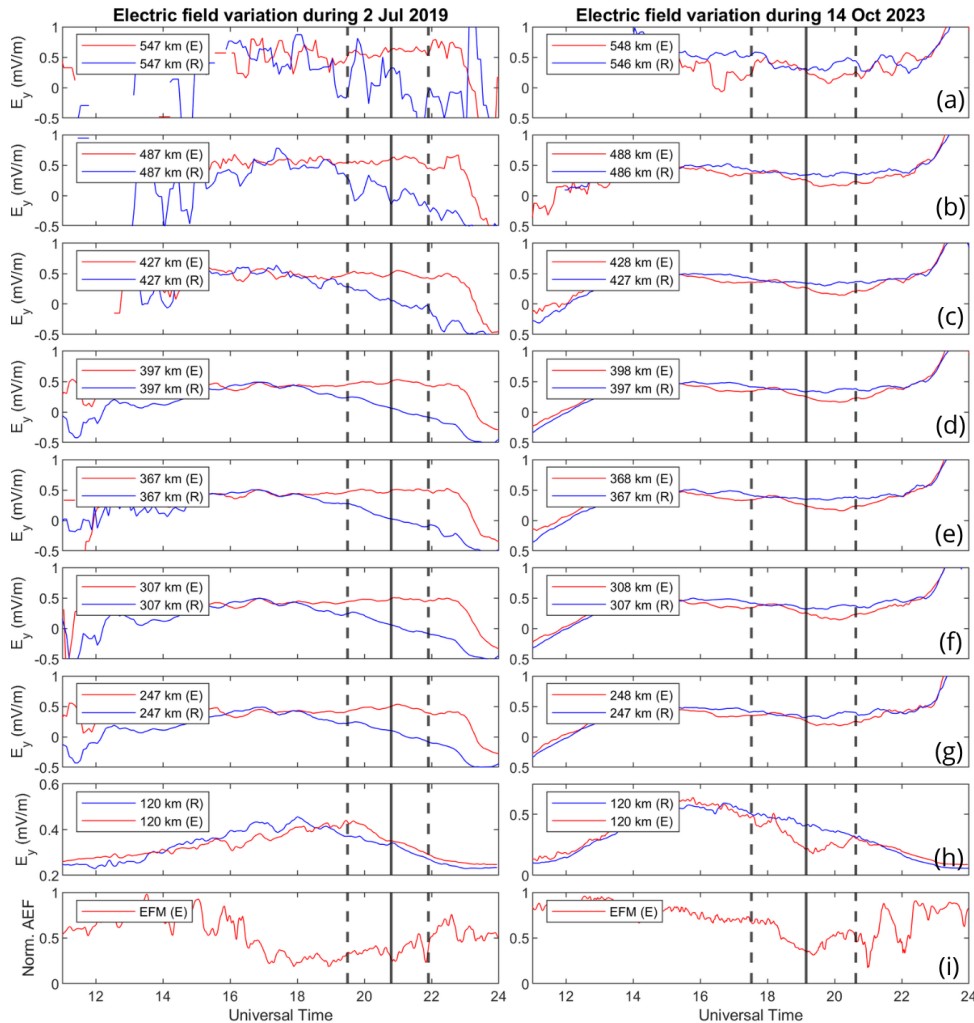

**Figure 7: Electric field variation at different heights during both solar eclipses estimated from zonal drift velocity and obtained from EFM-100 (last row). Maximum obscuration (vertical black continuous line), eclipse onset and eclipse end (vertical black dashed line) are indicated.**



Figure 7 shows the zonal component of the electric field (positive eastwards) derived from the measured drifts of the Jicamarca ISR (2019) and the JULIA-MP radar (2023), as well as that derived from magnetometers at 120 km and normalized AEF (using Tacza et al., 2018 technique). A 5-point moving mean was applied to every time series. For the 2019 case, a decrease near the maximum obscuration time is observed at ground level and at 120 km, followed by an increase. This behavior is consistent with the photoionization dependence at lower altitudes in the ionosphere. However, higher altitudes show no decrease, which is consistent with the predominance of transport processes over solar dependence for this event (Bravo et al., 2020; Jonah et al., 2020). On the other hand, the 2023 case shows a much clearer behavior with a decrease at all altitudes, with a quicker response at lower altitudes, which is consistent with the solar radiation dependence as the main ionization source below 200 km (pending citation).

## 4 Discussion

The results obtained from the atmospheric and ionospheric electric field measurements during the solar eclipses of 2 July 2019 and 14 October 2023 offer important insights into how such transient events affect Earth's electrodynamic environment at equatorial latitudes. However, several factors complicate the interpretation, particularly in the AEF data, which are discussed in the following sections.

In both eclipse events, a noticeable increase in AEF was observed near the time of maximum obscuration. However, the high day-to-day variability of AEF introduces significant uncertainty into this observation. This variability is a well-documented feature of surface-level electric field measurements, which are highly sensitive to local meteorological conditions such as wind, humidity, and cloud cover (Bennett, 2016; Velasquez et al., 2022). Similar fluctuations have been observed in previous eclipse studies, where distinguishing eclipse-induced signals from meteorological noise remains challenging (Silva et al., 2020).

In the 2019 eclipse, although no local meteorological data were available, satellite imagery from GOES-16 suggested relatively stable conditions. Compared to the FW-SC reference, the measured AEF reached approximately twice the typical daytime value, suggesting a potential eclipse-related enhancement. For the 2023 eclipse, both statistical and meteorological criteria were used. The FW-MC reference curve showed higher daytime AEF values than FW-SC and included nighttime negative excursions that were absent on the eclipse day. The solar radiation profile exhibited a Gaussian-like shape, with a distinct dip during the eclipse and no indication of cloud interference, suggesting minimal meteorological influence. Yet, the AEF values during the eclipse remained within the bounds of the fair-weather range, making it difficult to isolate eclipse effects from ambient variability. Although a short-lived AEF reduction during maximum obscuration appears consistent with previous observations (Anderson and Dolezalek, 1972; Tacza et al., 2016), the lack of persistent or pronounced deviations limits conclusive attribution.

In contrast, the ionospheric electric field (represented by the vertical E×B plasma drift) showed more robust and interpretable responses, though with differing behaviors in each event. During the 2 July 2019 eclipse, a significant increase in vertical drift was observed around the time of maximum obscuration, peaking approximately 25 m/s above the control day. This may reflect a transient intensification of equatorial electrojet-related processes or shifts in ionospheric conductivity gradients



caused by asymmetric EUV radiation reduction. Conversely, the 14 October 2023 event exhibited a marked suppression of vertical drift near eclipse maximum, followed by a delayed recovery. This behavior aligns with expected reductions in dynamo-driven electric fields due to diminished ion production in the E and F regions. Le et al. (2009) documented similar

drift suppressions during eclipses with high obscuration levels. The contrasting responses highlight the complexity of ionospheric electrodynamics under eclipse conditions, likely modulated by differences in local time, solar zenith angle, and background ionospheric states. Notably, zonal drift variations were less pronounced in both cases, suggesting that vertical drift is more sensitive to eclipse-induced perturbations at low latitudes.

The comparison with vertical drift estimates derived from magnetometer data further supports the radar-based findings.

Particularly during daytime hours (when magnetometer-derived estimates are more reliable) the consistency between Jicamarca radar observations and HUA-ARQ differential data reinforces the conclusion that ionospheric electric fields are affected by eclipses, most likely through changes in ionospheric conductivity and current systems.

The simultaneous consideration of AEF at the surface and E×B drift in the ionosphere offers a perspective on vertical electrodynamic coupling. Although establishing a direct one-to-one correspondence is challenging due to differing

sensitivities and spatial scales, the observed AEF enhancements near eclipse maxima loosely coincide with changes in vertical drift. This inverse or asynchronous behavior may indicate a redistribution of electric potential along the atmospheric column, as suggested by Martínez (2014). Additionally, some authors propose that gravity waves triggered by rapid solar obscuration may act as coupling mechanisms between the lower and upper atmosphere (Huba and Krall, 2013; Barad et al., 2022). While such mechanisms could have contributed to the observed ionospheric perturbations, further data are needed for

confirmation.

Vertical coupling of atmospheric electric fields is particularly complex at equatorial latitudes, where the geomagnetic configuration allows for more efficient transmission of electric field variations between atmospheric layers (St-Maurice et al., 2011). The global atmospheric electric circuit framework supports this idea, with the Earth's surface and ionosphere acting as equipotential boundaries connected by vertical currents regulated by atmospheric conductivity (Rycroft et al., 2000;

Tinsley, 2008). Mathematical simulations, such as Denisenko et al., (2018), reveal the intricate nature of equatorial electric fields, though with relatively small magnitudes, which makes difficult to observe how important is the contribution of the eclipse impact on this global circuit, even when solar eclipses can temporarily disrupt this system by reducing ionization and modifying conductivity profiles, thus altering current flows and electric field distributions. Even minor changes in solar forcing or conductivity at these latitudes can affect ionospheric electric fields (Forbes et al., 2000; St-Maurice et al., 2011).

Separately, theoretical models of lithosphere–atmosphere–ionosphere coupling have proposed that anomalous electric fields may propagate upward prior to seismic events, driven by vertical currents associated with radon emissions or aerosol charging (Namgaladze, 2013). However, these models often require unrealistically large currents to explain significant ionospheric anomalies under fair-weather conditions (Surkov and Pilipenko, 2024), and the observed AEF variations in this study remain within expected meteorological variability.

Given the amplitude of the observed AEF changes and the known influence of local weather, conclusive evidence for vertical electric coupling remains elusive. A more definitive assessment requires coordinated measurements of electric fields, atmospheric conductivity, and neutral dynamics across multiple altitudes.

The interpretation of our results is constrained by several limitations. First, incomplete data coverage (especially during the 2023 eclipse) hampered continuous monitoring. Second, the number of fair-weather control days was limited, particularly in October 2023, reducing statistical reliability. Third, the absence of direct atmospheric conductivity measurements prevented quantitative assessment of changes in ion production. Future studies should prioritize multi-instrument and multi-site campaigns, including vertical profiling of conductivity and neutral winds. Coordinated observations during future eclipses, especially at equatorial stations like Jicamarca, may help isolate electrodynamic signatures and advance our understanding of vertical coupling mechanisms across atmospheric layers.

## 5 Conclusions

The observations of atmospheric and ionospheric electric fields during the solar eclipses of 2 July 2019 and 14 October 2023 provide valuable insight into the electrodynamic response of the equatorial environment to transient solar forcing. Although the atmospheric electric field (AEF) exhibited apparent fluctuations near the time of maximum obscuration in both events, the high natural variability of surface-level measurements complicates any attempt at direct attribution. The amplitudes observed remained largely within the bounds of fair-weather variability, and the influence of local meteorological conditions could not be fully ruled out. In the case of the 2019 eclipse, a marked increase in AEF was noted; however, the absence of in situ meteorological data limits a conclusive interpretation. For the 2023 eclipse, a more comprehensive approach incorporating both statistical and meteorological reference curves confirmed internal consistency, yet again no unequivocal AEF response attributable to the eclipse could be established.

In contrast, the ionospheric response, quantified through E×B plasma drifts, revealed more distinct and interpretable changes. The 2019 event was characterized by a significant enhancement in vertical drift velocities during and following the period of maximum obscuration, suggesting possible modulation of equatorial electrojet dynamics or conductivity gradients. The 2023 eclipse exhibited a clear suppression of vertical drift, followed by a delayed recovery phase. These opposing responses underline the sensitivity of ionospheric electric fields to background ionospheric conditions, solar zenith angle, and local time, and are consistent with the expected impact of reduced solar ionization on the E and F region dynamo processes. Meanwhile, zonal drift variations were less clearly affected in both events, suggesting a dominant role of vertical electrodynamics in eclipse-related perturbations at low latitudes.

The temporal association between anomalies in the surface AEF and changes in ionospheric vertical drift supports the possibility of vertical electrodynamic coupling. However, this relationship appears to be neither immediate nor linear and may reflect complex altitude-dependent processes within the atmospheric column. The observations are broadly consistent with the global atmospheric electric circuit framework, in which vertical current systems link the Earth's surface to the ionosphere, modulated by atmospheric conductivity. Nevertheless, the amplitudes of the AEF variations observed in this



study remained modest, and the absence of direct conductivity profiles or neutral wind data prevents a definitive assessment of the coupling efficiency or mechanisms involved.

Limitations in data continuity, particularly during the 2023 eclipse, along with the scarcity of fair-weather control days and the lack of key atmospheric parameters, constrain the robustness of the present analysis. Future campaigns should prioritize coordinated, multi-instrument observations across different altitudes, including direct measurements of atmospheric conductivity and neutral dynamics. Such efforts will be essential to disentangle meteorological and eclipse-induced effects and to improve our understanding of vertical electrodynamic coupling at equatorial latitudes.

In summary, the response of the ionosphere to solar eclipses appears more pronounced and consistent than that of the surface atmospheric electric field, reinforcing the importance of upper atmospheric observations in eclipse studies. While surface electric field data may offer complementary information, their high sensitivity to local conditions makes them less reliable as standalone indicators of solar eclipse effects. Only through integrated observations from the ground to the ionosphere will it be possible to isolate and characterize the subtle and transient electrodynamic impacts of solar eclipses.


**Author contribution**

Conceptualization: MAB, OP, BAU, JJS, EAC, EDR, EMO; Methodology: MAB, AG, BAU, JHF, JJS; Software: AG, BAU, CES; Validation: AG, BAU, JJS; Formal analysis: AG, BAU, JHF, JJS; Investigation: MAB, OP, AOC, JHF, JEP, CES, PQ, EV; Resources: OP; Data curation: CES, AG, BAU, AOC, SMR, HCM, GEM; Writing – original draft: MAB,
OP, AG, BAU, JHF, JJS, JEP; Visualization: AG, BAU, EAC. JJS, JHF; Supervision: MAB, OP; Project administration: OP; Funding acquisition: OP, MAB.

**Funding**

This research did not receive external funding. The APC was funded by Universidad Peruana Unión and Universidad Adventista de Chile.

**Competing interests**

The authors declare that they have no conflict of interest.

**Acknowledgments**

We are grateful to Universidad Peruana Union (UPeU) and Universidad Adventista de Chile (UnACh) for their valuable support. This work was funded by the UPeU under the internal competition PICB-2024-01 "Research Projects in Basic
Sciences," approved under Resolution No. 2135-2024/UPeU-CU, registered under code PICB02. The UnACh received funding for the Regular Internal Project No. PI-251. The members of UPeU would like to thank Dr. Jose Tacza of the Mackenzie Center for Radio Astronomy and Astrophysics, Faculty of Engineering, Mackenzie Presbyterian University, São Paulo, Brazil, and the AFINSA network for installing the EFM sensor on campus. We express our gratitude to the Geophysical Institute of Peru (IGP) Huancayo campus, Engineer Luis Flores, the Huayao Observatory, Huachac, and



Engineer Luis Fernando Suárez for their valuable contribution and unconditional support in this research. The authors acknowledge the assistance of Sider.ai in the translation and preliminary review of the manuscript draft, with subsequent comprehensive manual scientific validation.

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
