# Peer review of "Observations of Electric Fields during two Partial Solar Eclipses at the Geomagnetic Equator"

_EGUsphere, 2025_

## Author Response (AR1)

Thank you very much for the review. Your review has significantly improved our publication, making it more in line with the instrument observations.

1.- Comment: The authors did not mention about the need of the coordinated observations of Atmospheric Electric field and ionospheric plasma drifts. For example, they could have mentioned about the electrical and electrodyamic coupling between the upper and the lower atmosphere in terms of the spaceweather and terrestrial weather investigations at an appropriate location in the manuscript.

R: We appreciate this insightful suggestion, which highlights an opportunity to strengthen the manuscript's framing. Indeed, the original introduction briefly noted the novelty of our coordinated AEF and ionospheric drift observations (e.g., abstract and pages 1-3), with references to vertical coupling (Tacza et al., 2016; St-Maurice et al., 2011). To address your point, we will add a dedicated, properly cited paragraph in the revised introduction explicitly discussing the need for such observations in investigating electrodynamic coupling between the upper and lower atmosphere, including linkages to space weather (e.g., solar forcing effects) and terrestrial weather (e.g., meteorological influences on conductivity). This will incorporate additional references, such as Rycroft et al. (2000) and Pulinets and Ouzounov (2011), to better contextualize the study's contributions.

2.- Comment: The two observations considered in the present study display contradictory characteristics. The cited references of the past studies, in the similar topic, also do not provide a conclusive result. A few show increase of AEF during the solar eclipse and another few suggest decrease in the same.

R: We fully acknowledge the contradictory nature of AEF responses to solar eclipses, as reflected in our results and the literature review (e.g., increases in Anderson and Dolezalek, 1972; Babakhanov et al., 2013; decreases in Kumar et al., 2013; negligible effects in Bennett, 2016). Our pragmatic approach was to highlight this complexity without selectively favoring citations or adjusting results to prior conclusions; instead, we emphasized the intricate connections between solar eclipse phenomena and AEF through two events with opposing ionospheric responses under comparable conditions. This opposition appropriately illustrates the vertical coupling between atmospheric layers, as discussed in Sections 4 and 5. To clarify this, we will revise the discussion section to more explicitly frame these contradictions as evidence of electrodynamic linkages, while calling for further studies to resolve inconsistencies.

3.-Comment: The AEF is mostly influenced by the meteorological parameters. To examine the role of the contribution of the signatures of the upper atmosphere over the AEF it is very important to prove that there is no role of the meteorological weather parameters on the AEF. The weather information provided in this paper does not suffice the need.

R: We agree that meteorological influences on AEF must be rigorously controlled to isolate upper atmospheric signatures, as noted in our methods and discussion (e.g., citing Velazquez et al., 2022). However, the absence of local meteorological data for 2019 limited our analysis compared to 2023; the nearest available station was approximately 200 km away, rendering it unrepresentative due to differing geography. We mitigated this by using GOES-16 satellite imagery to minimize uncertainty regarding adverse weather conditions that could affect local AEF. For 2023, wind data analysis confirmed no significant meteorological perturbations that would obscure potential eclipse responses. To enhance rigor, we will expand the methods section in the revision to detail these limitations and justify our fair-weather criteria, while recommending future integration of proximal meteorological stations for more comprehensive validation.

4.- Comment: In Fig. 3 the authors represent the diurnal variation of AEF on a reference day and on the eclipse day. The fluctuation of the data on the day of event is stronger during the hours other than the eclipse duration. Therefore the increase or decrease cannot be attributed to the event.

In Fig. 4 also there are noticeable fluctuations in the diurnal variation of the AEF. However, there is a noticeable change in the AEF after the onset of the eclipse. It is believed to be due to diminishing of the dynamo driven electric fields. To ratify this one needs complimentary experiments such as atmospheric electrical conductivity and air ions measurement. These two parameters are highly sensitive to the meteorological weather parameters. Therefore it is difficult to infer whether this change in AEF is due to the eclipse event.

R: We recognize that the observed data exhibit suboptimal amplitudes due to installation errors, as mentioned in the methodology. The applied normalization, similar to Tacza et al. (2018), facilitates a qualitative rather than quantitative analysis to mitigate this impact. For the 2023 event in Fig. 4, the AEF response aligns temporally with the eclipse duration, whereas post-event rapid fluctuations are shorter-lived, suggesting they should not be conflated. In contrast, the 2019 data in Fig. 3 show even greater variability, which we explicitly noted. Regarding complementary experiments (e.g., atmospheric electrical conductivity and air ion measurements), these were not feasible due to equipment limitations; we will emphasize this as a challenge for future comparable eclipse events in the revised discussion, while referencing models like Denisenko et al. (2018) to support potential dynamo-driven interpretations.

5.-Comment: In the conclusion the authors claim that the atmospheric and ionospheric electric fields during eclipse provides valuable insight into the electrodynamic responses to transient solar forcing, but the results and discussions are not enough to come to this conclusion.

R: We agree that conclusions must strictly align with the presented evidence, avoiding excessive claims. The main purpose of this paper was to present coordinated observations and acknowledge the complexity of analyzing AEF effects. To illustrate this challenge, we included two eclipse events that showed opposite ionospheric responses. In light of the reviewer's comment, we will revise the conclusions section, moderating the language: it will now state that ionospheric data provide clear evidence of electrodynamic responses, while AEF inferences are described as preliminary and pending multi-instrumental validation. This adjustment will ensure consistency between results and conclusions, improving the overall clarity of the manuscript.

6.- Comment: The figures should be labelled properly. Where ever multiple panels are presented they should be labelled as (a), (b), (c) etc followed by appropriate description.

R: We concur with this recommendation for improved clarity. In the revised version, we will label all multi-panel figures (e.g., Figs. 3-7) with (a), (b), etc., and provide detailed descriptions in the captions.

7.- Comment: The time in the X axis in local time is more appropriate than the UT.

R: We agree that local time enhances interpretability for equatorial studies. In the next version, we will update the x-axes in relevant figures (e.g., Figs. 2-7) to prioritize local time, with universal time noted for reference where necessary.